# Tracking the introduction and spread of SARS-CoV-2 in coastal Kenya

George Githinji [1,2✉], Zaydah R. de Laurent [1], Khadija Said Mohammed [1], Donwilliams O. Omuoyo[1], Peter M. Macharia [3], John M. Morobe [1], Edward Otieno [1], Samson M. Kinyanjui[1,4], Ambrose Agweyu[1], Eric Maitha[5], Ben Kitole[5], Thani Suleiman[6], Mohamed Mwakinangu[7], John Nyambu[8], John Otieno[9], Barke Salim[10], Kadondi Kasera[11], John Kiiru[11], Rashid Aman[11], Edwine Barasa[4,12], George Warimwe[1,4], Philip Bejon[1,4], Benjamin Tsofa[1], Lynette Isabella Ochola-Oyier[1], D. James Nokes [1,13] & Charles N. Agoti[1,14]

Genomic surveillance of SARS-CoV-2 is important for understanding both the evolution and the patterns of local and global transmission. Here, we generated 311 SARS-CoV-2 genomes from samples collected in coastal Kenya between 17th March and 31st July 2020. We estimated multiple independent SARS-CoV-2 introductions into the region were primarily of European origin, although introductions could have come through neighbouring countries. Lineage B.1 accounted for 74% of sequenced cases. Lineages A, B and B.4 were detected in screened individuals at the Kenya-Tanzania border or returning travellers. Though multiple lineages were introduced into coastal Kenya following the initial confirmed case, none showed extensive local expansion other than lineage B.1. International points of entry were important conduits of SARS-CoV-2 importations into coastal Kenya and early public health responses prevented established transmission of some lineages. Undetected introductions through points of entry including imports from elsewhere in the country gave rise to the local epidemic at the Kenyan coast.

---

[1] KEMRI-Wellcome Trust Research Programme, Kilifi, Kenya. [2] Department of Biochemistry and Biotechnology, Pwani University, Kilifi, Kenya. [3] Population Health Unit, KEMRI-Wellcome Trust Research Programme, Nairobi, Kenya. [4] Nuffield Department of Medicine, University of Oxford, Oxford, UK. [5] Department of Health, Kilifi, Kenya. [6] Department of Health, Mombasa, Kenya. [7] Department of Health, Kwale, Kenya. [8] Department of Health, Taita Taveta, Kenya. [9] Department of Health, Lamu, Kenya. [10] Department of Health, Tana River, Kenya. [11] Ministry of Health, Government of Kenya, Nairobi, Kenya. [12] Health Economics Research Unit, KEMRI-Wellcome Trust Research Programme, Nairobi, Kenya. [13] School of Life Sciences and Zeeman Institute for Systems Biology and Infectious Disease Epidemiology Research (SBIDER), University of Warwick, Coventry, UK. [14] School of Public Health, Pwani University, Kilifi, Kenya. ✉email: ggithinji@kemri-wellcome.org

Severe acute respiratory syndrome coronavirus 2 (SARS-CoV-2), the aetiological agent of coronavirus disease 2019 (COVID-19), was first reported and confirmed in Kenya on 13th March 2020[1]. This was shortly after the World Health Organisation (WHO) declared COVID-19 a pandemic on 11th March 2020. SARS-CoV-2 outbreaks had already been confirmed in many parts of Asia, Europe and North America following detection in Wuhan City, China, in late December 2019[2]. Within a month of the first confirmed case, Kenya put in place COVID-19 containment measures including closure of international borders, a dusk to dawn curfew, closure of all universities and schools, restaurants, bars and nightclubs, and religious meetings (churches, mosques and others). Meetings and social gatherings with more than 15 people were banned. Movement into or out of areas that were considered epidemic hotspots became restricted, and the government introduced strict quarantine procedures and isolation of infected individuals. Despite these public health measures, the number of SARS-CoV-2 cases increased steadily across the country implying already established local transmission[3,4]. By 31st July 2020, Kenya had reported 20,636 laboratory PCR confirmed SARS-CoV-2 cases and 341 COVID-19 associated deaths[3]. SARS-CoV-2 seroprevalence data collected from the same period of time provided evidence that the number of exposed individuals in Kenya is likely to have been higher than the RT-PCR confirmed case reports[5].

The Kenyan coast is an international tourism hub and a major gateway for the East and Central Africa region. Although the government introduced screening at points of entry (PoE),

SARS-CoV-2 infections may have been imported into the region by persons coming through the multiple points of entry and including by road from Nairobi (Fig. 1a). The region has multiple international ports of entry by air (Mombasa, Malindi, and Lamu airports); land borders with Tanzania; and a seaport in Mombasa (Fig. 1b). Mombasa, Kenya's second largest city, emerged as one of the epicentres of the early wave of SARS-CoV-2 infections[4]. Mombasa county has a population of over 1 million (2019 census) distributed across seven administrative sub-counties, and reported more infections than the other, more rural, Coastal counties (i.e., Kwale, Taita Taveta, Kilifi, Tana River, and Lamu). Within Mombasa county, Mvita sub-county reported the largest number of cases (Fig. 1c).

Information on the early importation and spread of SARS-CoV-2 in Kenya is important in assessing the effectiveness of the early interventions, and designing additional COVID-19 control measures including vaccination programmes in Kenya and the East Africa region. A number of large-scale investigations have been described[6–8], and there are a limited number of studies and datasets available from East Africa[9], West Africa[10], and South Africa[11]. To date, there has not been any report on SARS-CoV-2 genomic epidemiology in Kenya to inform on the early introductions and spread of the virus locally. Here, we sequenced and analysed 406 RT-PCR positive samples collected between March and 31st July 2020 at the Kenyan Coast from which we obtained 311 sequences suitable for phylogenetic analysis to provide the first large-scale genomic epidemiology study of SARS-CoV-2 from the region.

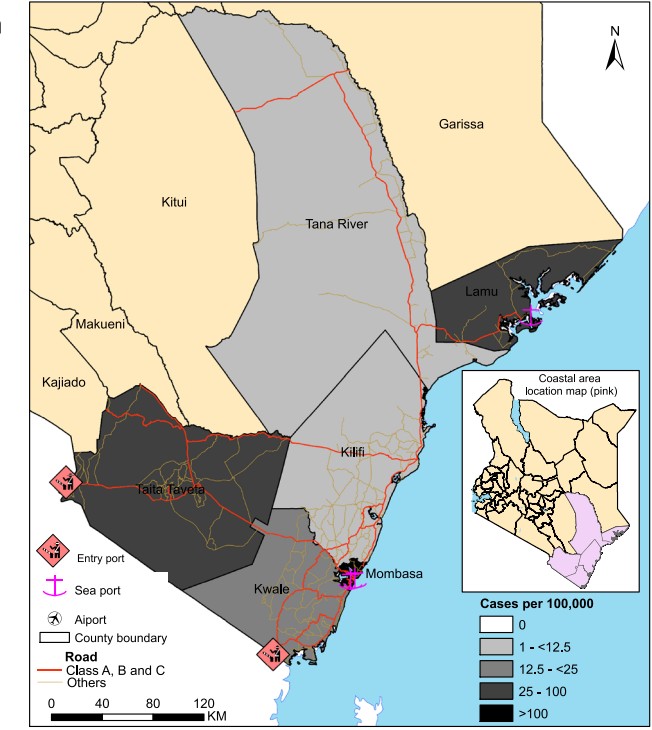
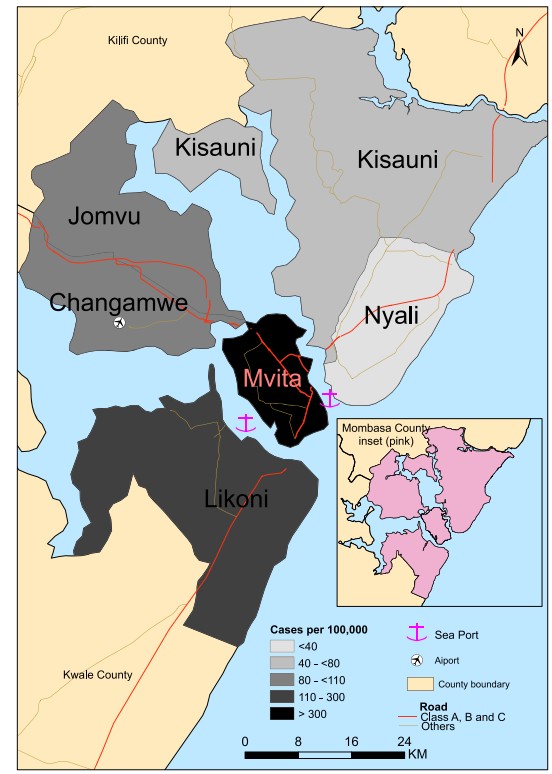

**Fig. 1 The geographical spread of SARS-CoV-2 at the Kenyan Coast. a** A geographical map of Kenya showing the main administrative counties of Kwale, Mombasa, Taita Taveta, Kilifi, Tana River and Lamu and together comprise the coastal region. The total number of confirmed SARS-CoV-2 positive cases per hundred thousand across the coast as at of 31st July 2020. The colour intensity is relative to the number of cases that were confirmed in the respective counties and overlaid on the major transportation infrastructure and hubs including road network, airport, seaport, and border or international entry points. The cases detected in Taita Taveta were largely from the One Stop Border Post at Taveta/Holili crossing point between Southern Kenya and Northern Tanzania. **b** A map of Mombasa county showing the spatial distribution of RT-PCR confirmed SARS-CoV-2 cases per 100,000. Mvita sub-county had the largest number of cases.

## Results

**SARS-CoV-2 testing and sequencing on the Kenyan Coast.**
SARS-CoV-2 RT-PCR testing was set up in mid-March 2020. This became a government designated testing centre to support the Department of Public Health Rapid Response Teams (RRTs) from the six coastal counties namely, Kilifi, Kwale, Lamu, Mombasa, Taita Taveta, and Tana River (Fig. 1). The testing criteria were dependant on the guidelines provided by the Ministry of Health (MoH), and we split the period into four phases (Supplemetary Table 1) depending on the number of cases that were reported in the country. By 31st July 2020, 1997 out of 37,925 tested samples were confirmed RT-PCR positive across the coastal counties (Figs. 1 and 2). Between 12th March and 31st July 2020, RRTs obtained samples from repatriated citizens including among individuals arriving at ports of entry, and from persons presenting at major hospitals with symptoms consistent with SARS-CoV-2 infection, contacts of confirmed cases and targeted testing of residents in Mombasa.

We analysed sequence data from 406 SARS-CoV-2 RT-PCR positive samples collected between 17th March 2020 and 31st July 2020 representing <1% of cases identified in our laboratory (Fig. 2). We utilised a total of 311 (Mombasa ($n = 227$), Kwale ($n = 29$), Taita Taveta ($n = 24$), Kilifi ($n = 13$), Lamu ($n = 14$) and Tana River ($n = 4$)) for phylogenetic analysis. The sequences were obtained after a conservative consensus genome building procedure followed by a thorough quality control to ensure not only high genome coverage sequences were included in subsequent analysis (80% completeness), but also provide confidence in the nucleotide sequences. These data included follow-up samples from 17 individuals (Supplementary Fig. 1).

Mombasa experienced the largest number of cases, while the counties of Lamu and Tana River did not report positive cases until 27th June 2020 and 17th July 2020, respectively. A total of 481 and 221 tests were carried out in both Lamu and Tana River between March and July.

**Demographic characteristics.** The median age of the individuals with sequenced samples across all the counties was 39 years (range 1–85 years), 66% of whom were male ($n = 268$), and more than half were asymptomatic (Table 1 and Supplementary Table 2). The sequenced cases from Taita Taveta were all from male individuals aged between 20 and 60 years and were all detected at the border point, suggesting that these infections may have been acquired outside Kenya (Supplementary Table 2). A number of the sequenced samples were from individuals with a history of recent travel ($n = 109$ (26%)), which was defined as international travel within last 2 weeks or were sampled at the port of entry ($n = 67$ (16%)) into Kenya (Table 1 and Supplementary Table 2). Approximately 41.3% of infected individuals had no history of travel outside their localities (Table 1 and Supplementary Table 2) providing evidence for established local transmission during this wave of the pandemic.

**Circulating lineages between March and July 2020 in coastal Kenya.** Overall, we observed multiple lineages across the coastal counties (Mombasa ($n = 20$), Kwale ($n = 7$), Taita Taveta ($n = 6$), Kilifi ($n = 6$), Tana River ($n = 1$) and Lamu ($n = 2$)). Mombasa, Kwale and Taita Taveta have multiple ports of entry into Kenya (Fig. 1b, c).

The dominant lineage at the coastal region was the B.1 lineage which was first sampled in March 2020 (Table 2, Supplementary Table 3 and Fig. 3a). The B.1. lineage was characteristic of outbreaks in Italy and the United Kingdom and was observed among several additional samples from outbreaks in Europe, which later on spread to the rest of the world[12]. In coastal Kenya, B.1 was detectable among early cases from samples collected in March 2020 and it continued to dominate the cases in the coastal region in the course of April–July 2020 (Fig. 3 and Supplementary Fig. 2). This lineage expanded rapidly and provided evidence of local transmission (Fig. 3) based on the phylogenetic inference

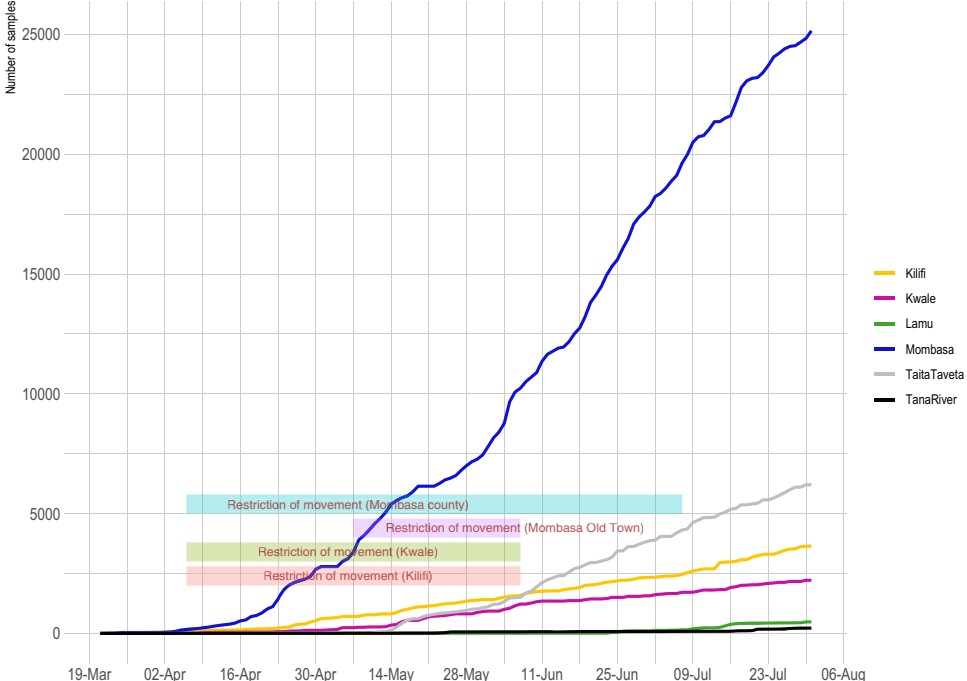

**Fig. 2 Testing of SARS-CoV-2 cases at the Kenya Coast.** The cumulative number of SARS-CoV-2 positives cases that were confirmed from each of the six counties at the Kenyan coast are represented by the lines. The horizontal bars represent the county specific public health interventions that were undertaken during the study period and early in the epidemic period. The length of the bars corresponds to the time duration for each respective intervention.

**Table 1 Demographic characteristics of SARS-CoV-2 positive samples collected between 17th March and 31st July 2020 (n = 406) from coastal Kenya.**

| | Mombasa (n = 287) | Other coastal counties (n = 119) | Total (n = 406) |
|---|---|---|---|
| *Sex* | | | |
| Female | 89 (31.0%) | 21 (17.6%) | 110 (27.1%) |
| Male | 176 (61.3%) | 92 (77.3%) | 268 (66.0%) |
| Unknown | 22 (7.7%) | 6 (5.0%) | 28 (6.9%) |
| *Age* | | | |
| Mean (SD) | 40.7 (16.2) | 37.6 (11.8) | 39.8 (15.0) |
| Median | 40.0 | 35.0 | 39.0 |
| [Min, Max] | [1.00, 85.0] | [12.0, 75.0] | [1.00, 85.0] |
| Missing | 14 (4.9%) | 2 (1.7%) | 16 (3.9%) |
| *Age category* | | | |
| 0–9 | 10 (3.5%) | 0 (0%) | 10 (2.5%) |
| 10–19 | 10 (3.5%) | 2 (1.7%) | 12 (3.0%) |
| 20–29 | 41 (14.3%) | 26 (21.8%) | 67 (16.5%) |
| 30–39 | 71 (24.7%) | 43 (36.1%) | 114 (28.1%) |
| 40–49 | 55 (19.2%) | 27 (22.7%) | 82 (20.2%) |
| 50–59 | 53 (18.5%) | 13 (10.9%) | 66 (16.3%) |
| 60–69 | 23 (8.0%) | 2 (1.7%) | 25 (6.2%) |
| 70–79 | 6 (2.1%) | 4 (3.4%) | 10 (2.5%) |
| 80–89 | 4 (1.4%) | 0 (0%) | 4 (1.0%) |
| Missing | 14 (4.9%) | 2 (1.7%) | 16 (3.9%) |
| *Travel information* | | | |
| Border | 19 (6.6%) | 48 (40.3%) | 67 (16.5%) |
| Local | 141 (49.1%) | 27 (22.7%) | 168 (41.4%) |
| Travel associated | 13 (4.5%) | 29 (24.4%) | 42 (10.3%) |
| Unknown | 114 (39.7%) | 15 (12.6%) | 129 (31.8%) |
| *Symptoms* | | | |
| Asymptomatic | 155 (54.0%) | 76 (63.9%) | 231 (56.9%) |
| Symptomatic | 43 (15.0%) | 7 (5.9%) | 50 (12.3%) |
| Unknown | 89 (31.0%) | 36 (30.3%) | 125 (30.8%) |

The case history demographic characteristic was derived from both self-reported travel history and presentation at a border point. Local case-history refers to individuals that did not report a history of travel and were not screened at a port of entry. Individuals with missing case histories were labelled as unknown.

**Table 2 A summary table of early SARS-CoV-2 introductions into the Kenyan coast stratified based on the most frequent lineages.**

| Lineage | Introductions (approximate) | Percentage of sequences | Date of initial case | Source | Symptoms | Comments |
|---|---|---|---|---|---|---|
| A | 15 | 6.14 | 4th April 2020 | Household/After entry point | Yes | Returning from Dubai |
| B | 5 | 1.96 | 13th April 2020 | Household/After entry point | No | Returning from Dubai |
| B.1 | 8 | 74.44 | 17th March 2020 | Local/surveillance | Yes | History of travel |
| B.1.1.119 | 4 | 0.98 | 13th May 2020 | Border/Entry | No | Travel history to Tanzania |
| B.1.1.33 | 3 | 4.42 | 11th May 2020 | Targeted testing | No | Detected during target testing |
| B.4 | 2 | 0.98 | 4th April 2020 | Border/Entry point | No | Two sampled at port of entry, one with travel history to Zambia |

Each row represents a major lineage and the date it was first observed from the sequence data and the suspected entry route (source) into the region, and whether the introductory case was associated with symptoms at the time of sampling. A detailed breakdown of all the lineages is shown in Supplementary Table 3.

and ancestral reconstruction of internal nodes using country as a discrete character trait. The node nearest to the root supported evidence for an introduction from Italy based on a large set of contextual sequences. To estimate the number of introductions of this lineage, we focused on the period between March and 30th April by inferred ancestry of internal nodes. Our data supports evidence for multiple introductions on this lineage into coastal Kenya from Europe. In addition, it is plausible that a single introduction could have been undergoing transmission in early March prior to the collection of the first confirmed case from the coast (Supplementary Fig. 7).

Lineage A was the second most frequent lineage particularly among individuals sampled at the ports of entry (PoE) based in Kwale and Taita Taveta counties (Fig. 3, Table 2 and Supplementary Table 4) or among individuals with a history of travel (n = 8) to Tanzania (Table 1). We described at least 15 independent introductions of this lineage into the coast region (Table 2 and Supplementary Fig. 6). The first lineage A sample was detected in Mombasa from samples collected on the 4th of April (C314, GenBank accession MW751133—detected at the border and travel associated sample C293, GenBank accession MW751131). These two samples and in addition to sample C571 (GenBank accession MW751146), differed by one mutation from the reference and formed a single phylogenetic clade, whose ancestral date was inferred as 26th March 2020 (date confidence interval 18th March–1st April 2020). Samples C7603, C7605,

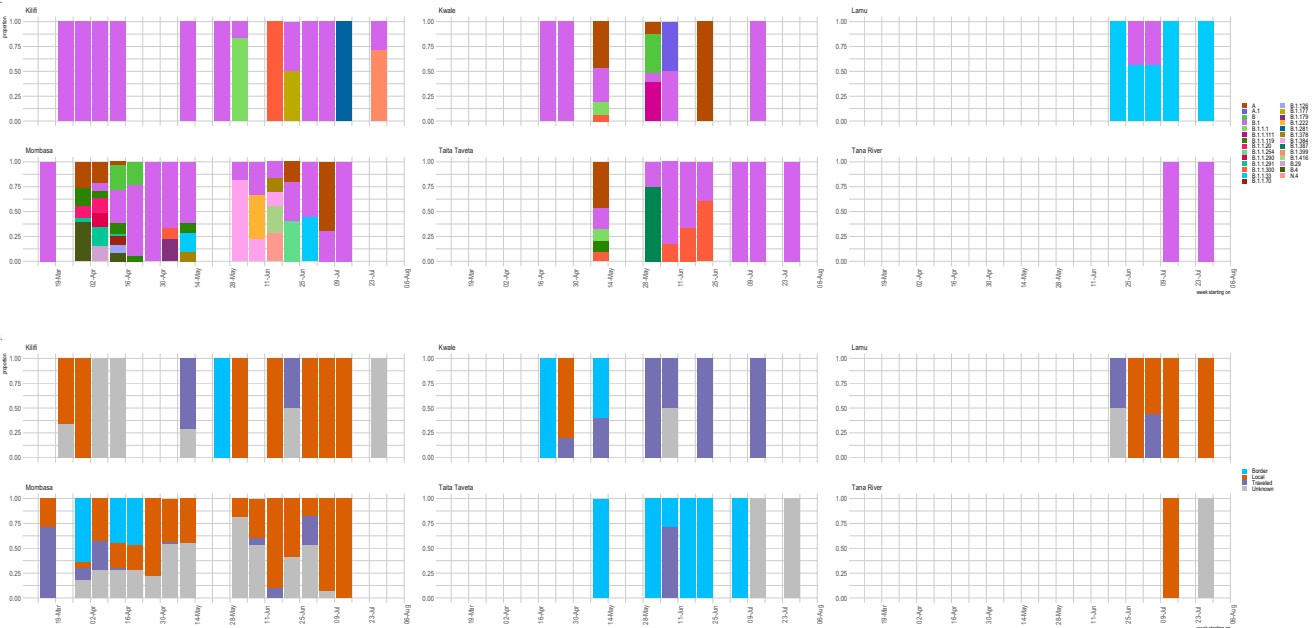

**Fig. 3 Sequence and genetic diversity of sequences during the early phase of the epidemic in coastal Kenya. a** A bar graph showing the proportion of assigned lineages to 406 SARS-CoV-2 sequences collected from the coast between 17th March to 31st July 2020 and stratified by county. The colours represent the lineages that were identified. **b** A bar graph showing the proportion of 406 SARS-CoV-2 sequences from the coastal region and the associated overall epidemiological source information aggregated by week and stratified by county. The colours represent whether a confirmed infection was detected at a border point of entry, travel associated or was a local case.

C7866 and C14075 (GenBank accession MW751307, MW751308, MW751311 and MW751362) also formed a single cluster but comprised of multiple introductions given that they were detected at the Lunga Lunga PoE and collected on different time points (Supplementary Fig. 7). Two additional samples collected on the 14th May 2020 support a common source of infection but comprised two separate introductions. Our ancestral reconstruction provides evidence that these multiple introductions of lineage A could be of Asian origin (Supplementary Fig. 6). We identified a single lineage A.1 sample, which could be an indicator for transmission of this lineage in the region. The paucity of genomic data from the region limits our description and understanding of the local dynamics of transmission. Nonetheless by focussing our analysis on the 32 samples with known or documented history of travel (Supplementary Table 4), we show that a number of cases detected at the borders points of Kwale ($n = 13$) and Taita Taveta ($n = 18$) were flagged as asymptomatic ($n = 22$). This provides further evidence that these could be sources of virus introduction into the region. In addition, the data provides a glimpse of circulating lineages in Tanzania despite paucity of genomes among the early cases from this region. Interestingly, despite potential introduction of lineage A cases in Kenya, infections in coastal Kenya were driven by lineage B.1.

Lineage B.1.33 was dected in Lamu and comprised the earliest number of cases from Lamu. ($n = 16$). Two cases were reported in Mombasa from samples collected on 11th May 2020 and 29th June 2020, respectively (Supplementary Fig. 2). The sample collected on 11th May 2020 clustered separately from the rest of the samples and could have comprised a separate introduction in Mombasa. The sample collected from Mombasa on the 29th June 2020 was part of a phylogenetic clade with the Lamu samples (Supplementary Fig. 2). Analysis of the ancestral nodes provide evidence for an ongoing epidemic in Lamu prior to the date of the first sequenced sample. Lamu is the only county we observed an

epidemic from a SARS-CoV-2 lineage associated with sequences from South America. (Supplementary Figs. 5 and 7).

The B.4 cases were sampled in Mombasa from four individuals, three of whom were screened at a point of entry (with a travel history from Lusaka, Zambia) and one individual who had no travel history. The B.4 lineage has been previously reported amongst individuals with a travel history to Iran[12]. The rest of the lineages were observed less frequently.

**SARS-CoV-2 sequence diversity**. A majority of the sequences were characterised by between 4 and 16 nucleotide substitutions (Supplementary Fig. 3) relative to the Wuhan reference sequence (NC_045512.2) and many of the coastal sequences contained a mutation at position A23403G (S:D614G) in addition to three other mutations (P314L, P970L, R203K and G204R). The D614G mutation arose early during the pandemic[13] and has become the dominant variant across the globe[14].

A total of 17 individuals placed in quarantine facilities were sampled repeatedly (Supplementary Fig. 1). The most frequent lineage among the individuals was the B.1 ($n = 15$), one individual was infected by lineage A and another one with lineage B.1.1. Infections from four individuals (R_10, R_9, R_5 and R_1) showed lineage discordance over the period of infection (Supplementary Fig. 1). Sequences in individual R_1 were characterised by amplicon drop-offs, which could have resulted in the observed differences, for example amino acid mutations at position H712Y and P1302L in ORF1a were missing at the respective position in each sequence due to missing sequence information (Supplementary Fig. 4). Sequence C677 (GenBank accession MW751153) collected at a later date appear to have acquired an additional synonymous mutation at position C28232T. On the larger phylogenetic grouping both sequences belonged to clade 20B and were characterised by mutations at C241T, C3037T, C14408T, A23403G (D614G) and G28883C that

resulted in an amino acid change (G204R) in the N protein. Three samples were collected from individual R_5 (7th, 13th and 18th April) from a 53 year male. The sequences appeared to fall into three different lineages (Supplementary Fig. 1) and spread out across three phylogenetic clades. Each of the sequences contained 7, 5 and 4 mutations away from the reference. The first sequence was characterised by more N's relative to the rest, which were more than 28,500 bases complete. The first two sequences contained D614G mutation on the spike and which was missing from the sequence that was collected on the 18th April. Nonetheless these data suggest that the sequences were very divergent and raises questions on multiple virus infections. Individual R_9 had two samples collected on the 13th and 22nd April 2020. The sequences fell into lineage B.1, B and clade 20A and 19A, respectively. The sequence collected at a later date was characterised with more incomplete genomic data that could have obscured two mutations that were observed at positions C14408T (ORF1b:P314L) and G29742A. Both sequences were characterised by the D614G mutation on the spike protein. The later sequence contained an additional mutation at position G24368T (S: D936Y). The sequences from the 4th individual R_10 were characterised by shared mutations at positions C3037T, C14408T (ORF1b:P314L) and A23403G (D614G). In addition, the sequence collected at a later date contained additional mutations at positions C18981T, G26062T (ORF3a:G224C) and G28883C (N:G204R). We could not ascertain whether mutation at position C241T was present in the subsequence sequence due to missing information. Samples collected from individual R_18 were collected about 15 days apart. The sample collected at a later time-point contained only 2 mutations both of which occurred on the spike protein at position G22017T (S:W152L) and G24378T (S:S939F).

**Phylogenetic clustering of sequences from coastal Kenya.** We estimated the number of independent introductions of SARS-CoV-2 into coastal Kenya by phylogenetic analysis of the local sequences and by sampling from a large contextual global dataset ($n = 2077$) of sequences collected before July 31st, 2020. Sequences from Kenya were distributed across all the main phylogenetic tree (Supplementary Fig. 5). The phylogenetic interspersing of the local viruses within the global sample provided evidence of multiple viral introductions into the local population (Table 2, Fig. 4 and Supplementary Fig. 6), resulting in at least 34 independent SARS-CoV-2 introductions into coastal Kenya. The observed local clusters were varied in both size and diversity, and some were suspected to have comprised further multiple independent introductions of closely related sequence variants. The results from the temporal clustering (Fig. 4) were consistent with known case history and with our expectations of SARS-CoV-2 importations into the region (Supplementary Figs. 7 and 8).

**Discussion**

In this work, we used genomic surveillance to elucidate the origins and transmission of SARS-CoV-2 in the Kenyan coastal population. Our genomic sequence and epidemiological data reveal an upper-bound number of 34 and 37 introductions of SARS-CoV-2 based on a computational approach (Supplementary Fig. 5) and a count of introduced lineages (Table 2 and Supplementary Table 3 and Supplementary Fig. 2) in the Coastal region of Kenya. We found evidence of established local transmission of the B.1 lineage in the Coastal region particularly in Mombasa county (Although several major global viral lineages were detected in returning citizens or international travellers, at the entry points or within the country, few resulted in significant

outbreaks (Fig. 4). A possible explanation for this is that the government action of enhanced border control and screening, quarantine, isolation including contact tracing of positive cases were effective in mitigating transmission. In addition, restrictions in international air-travel into the country reduced the number of potential introductions. This is corroborated by data showing low $R_0$ values in the early phase of the epidemic[4]. Differences in the transmissibility of the different lineages is possible[13,15] but in our case the predominance of B.1 in local transmission can be explained by the fact that most introductions were of B.1 lineage. The detection of lineage (B.1.1.33) in a sample that was collected during targeted testing (11th May 2020) in Mombasa city could be indicative of cryptic transmission within the region in spite of the intervention, but also could be failure to detect this lineage at the port of entry due to sampling limitations. Interestingly, the first and subsequent cases that were observed in Lamu comprised of the B.1.1.33 lineage and could be explained by a potential founder effect. Lamu experienced SARS-CoV-2 introductions 3 months into the Mombasa infections and following easing of travel restrictions between counties.

Early COVID-19 control strategies by the Kenyan government were geared towards preventing establishment of local community transmission. These policies appear to have been at least partially successful in that most of the introductions were not associated with subsequent established transmission. Nevertheless, a minority of introductions did go on to establish sustained local transmission and gave rise to the Kenya epidemic despite reduced virus lineage introductions. This underlines the severe challenge to the strategy that aimed at preventing the introduction of virus as any cases escaping the net had potential to establish community spread.

A total of 67 sequences were collected from individuals who presented at a border point of entry with majority having documented travel history in Tanzania ($n = 30$), Burundi ($n = 1$) or Zambia ($n = 1$). This provides evidence for a number of introductions through the PoE of Taita Taveta and Kwale. The lack of sequenced genomes from other counties in sub-Saharan Africa during this region complicates the inference of locations of introduction into Kenya. In neighbouring Uganda, international truck drivers including those from Kenya were identified as common sources of the infection[16,17]. This partly led to targeted testing of truck drivers in Kenya and a requirement of a "COVID-19 free certificate" at points of entry with neighbouring countries. Genomic analysis of SARS-CoV-2 cases in Uganda detected lineages similar to those reported in our study (i.e., A, B, B.1, B.1.1, B.1.1.1 and B.4)[9]. These findings emphasises the need for common or synergistic COVID-19 control strategies across East Africa to control the pandemic effectively. Globally, the B.1 lineage was a major European lineage first identified on February 15th, 2020. It will be interesting to continue evaluating the local spread and sustainability of this lineage in Africa as the epidemic evolves.

In our assessment of fresh introductions, we argue that a lineage defined an introduction if it was observed during the early phase of the epidemic and if the individual had a history of travel. For example, lineages detected between March and April could be regarded as potential introductions. During this period, the virus was more or less the same (2–4 mutations from the reference sequence) and therefore, we lose resolution on the number of introductions when a lineage is imported multiple times. We attempt to address this by looking at travel history of the cases and suggest that if these individuals arrived in the region at different time-points particularly, where such a sequence clusters more closely related with global sequence than local sequences, then this indicates a potential introduction. We use this approach to estimate at least 24 lower bound introductions. This approach

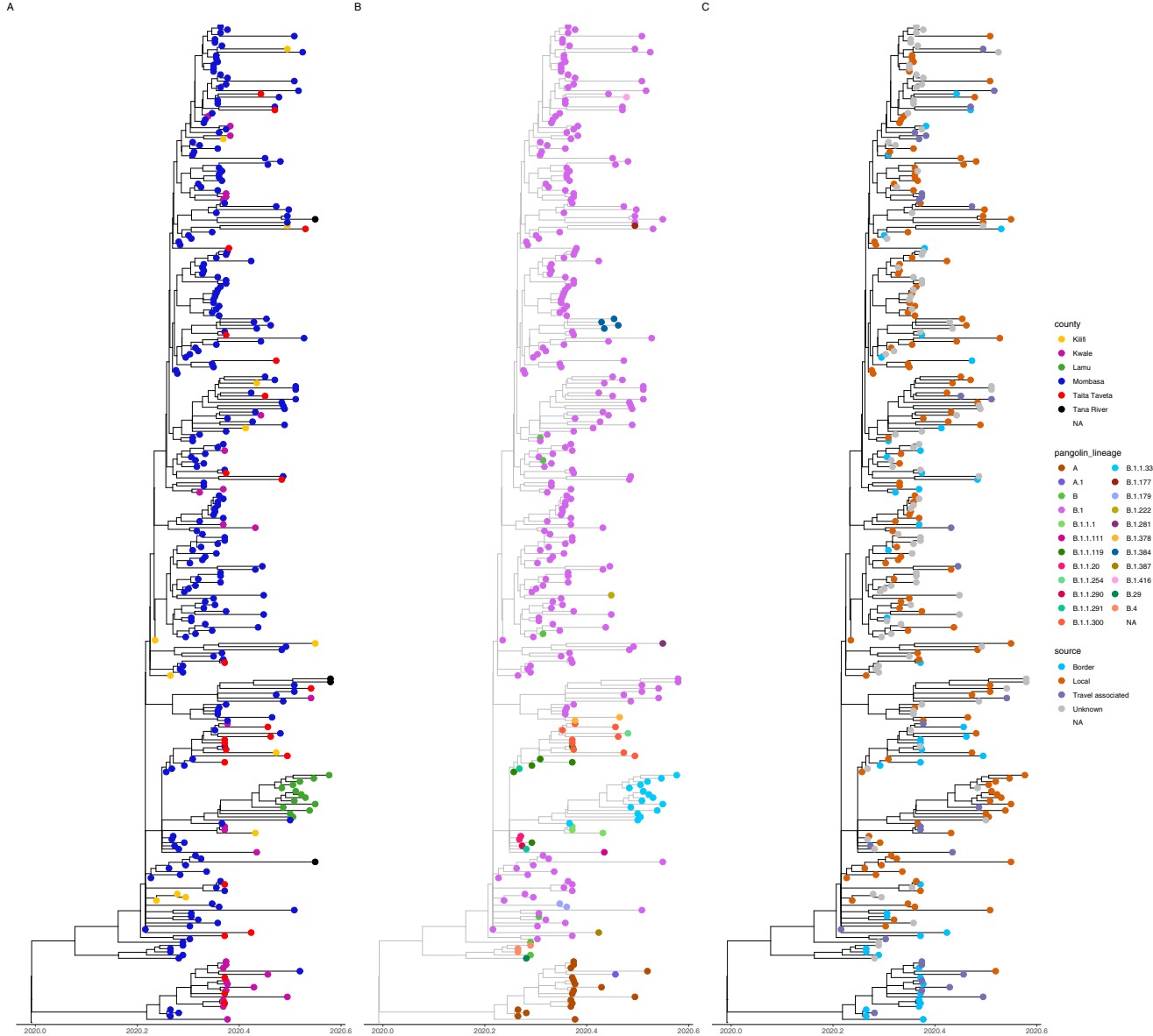

**Fig. 4 A time resolved phylogenetic tree of sequences collected from the coast Kenya. a** Geographical location. The tip colour represents samples collected from each of the coastal counties. **b** A time-resolved tree showing the Pangolin assigned lineage for each of the coastal sequences. **c** A time-resolved tree showing the samples stratified based on the travel history or detection at a border point of entry.

is limited by the definition of a lineage and in addition implies the number and sampling criteria of sequences from the global dataset is important and influences the number of estimated introductions. We apply a computational method implemented in PastML and estimate at least 34 introductions could have occurred. and both approaches provide evidence that the introductions were of European origin.

There are a number of limitations in our study for example the lack of complete epidemiological data for a number of demographic characteristics (Table 1 and Supplementary Table 2). In addition, a number of genomes ($n = 91$) were incomplete genomes due to amplicon dropouts or from samples with relatively low viral load. Obtaining representative samples was a challenge given that the testing strategy was not systematic, and testing guidelines were repeatedly revised as the epidemic evolved (Supplementary Table 2). This might have resulted in bias in the observed lineages. Although we present a substantial number of genomes from a single region within East-Africa, these data may

not generalise across the region. Nevertheless, our data provides a background for monitoring the progress of the SARS-CoV-2 epidemic in Kenya and a platform for continued surveillance and to build evidence for routes of transmission, sustained spread, effectiveness of control interventions and evolution of the virus within the region and at varying spatial resolutions.

Overall, our data provides evidence for multiple introductions of SARS-CoV-2 into the coast region and evidence for limited transmission of lineages that came in through the border points for example lineage A and B.4. Screening at the border and early surveillance efforts with contact tracing or isolation and quarantine of identified cases appears to have had considerable success in preventing the majority of introductions from leading to further transmission. Nevertheless, we hypothesise that a number of cases went undetected (Fig. 1a) and seeded local community transmission. This is particularly clear when looking at the cases attributed to lineage B.1 in Mombasa (Figs. 3 and 4).

Our analysis revealed the extent of imported SARS-CoV-2 genomic diversity that was seeded in local communities and the corresponding routes of novel introductions. We conclude that majority of the introductions came from Europe (Supplementary Fig. 5). Most of the introductions detected at border points did not result in community transmission, which reinforces the importance of SARS-CoV-2 testing at border crossing points and strict quarantining of positive cases during the early phases of an epidemic when the goal was to prevent transmission becoming established, but also the fragility of such methods when cases go undetected. These data have assisted to evaluate the effectiveness of government public health interventions and were rapidly shared with the Kenya Ministry of Health to inform on relevant public health response.

## Methods

**Ethics statement**. Samples were collected under the Kenya Ministry of Health (MoH) protocols as part of the national response to the COVID-19 pandemic. The whole genome sequencing study protocol was reviewed and approved by the Scientific and Ethics Review Committee (SERU) residing at the Kenya Medical Research Institute (KEMRI) headquarters in Nairobi (SERU # 4035).

**Study site and population**. The sequenced specimens were collected by the Rapid Response Teams (RRTs) from six counties in coastal Kenya namely, Kilifi, Mombasa, Kwale, Taita Taveta, Tana River and Lamu, between 17th March and 31st July 2020. Sample availability depended on the SARS-CoV-2 testing eligibility criteria and was revised several times by the Ministry of Health during the analysis period, which was roughly divided into four phases (Supplementary Table 1) for convenience. Individuals were eligible for testing during the first phase if they showed specific respiratory illness symptoms (cough, fever or difficulty in breathing), and had a recent history of international travel or were listed as close contact of a confirmed case. This phase was focused more on returning travellers from affected countries (mostly China, Iran, Italy and USA). There was limited testing centres, and no positive cases or genomes were obtained during this phase. The second phase started after the first SARS-CoV-2 confirmed case in Kenya (13th March 2020). In addition to guidelines that were provided during the first phase, the government directed that everyone arriving by air from international visits should proceed to a quarantine facility for 2 weeks. An individual would be released after a negative RT-PCR test and which marked the end of the quarantine period. The third phase began after increased cases from non-travellers which raised suspicion of community transmission in parts of Mombasa and Nairobi. The government recommended and rolled out targeted large-scale testing of workers at the Kenya Ports Authority (KPA) in Mombasa, and the general public in the Mvita sub-county (also referred to as the Old town or Mombasa island) (Fig. 1). The fourth phase began after increased reports of SARS-CoV-2 infections among truck drivers and which was corroborated by border surveillance teams in Uganda. Relevant to this study, the Kenya Ministry of Health (MoH) ordered targeted testing for truck drivers including those entering Kenya from Tanzania through the Lunga Lunga border (Kwale county) and Holili border point (Taita Taveta county). Our laboratory testing efforts throughout this period included patients with respiratory symptoms consistent with COVID-19 presenting to Mombasa hospitals, namely the Aga Khan Hospital, Mombasa Hospital, Coast Provincial Referral and Teaching Hospital (CPRTH) and The Premier Hospital.

**SARS-CoV-2 diagnosis at KWTRP**. The RRTs collected nasopharyngeal (NP) and oropharyngeal (OP) swabs into a single tube and transported them to the KEMRI-Wellcome Trust Research Programme (KWTRP) for SARS-CoV-2 diagnosis. The laboratory diagnostic protocol for SARS-CoV-2 at KWTRP has been described elsewhere[8,10] Briefly, the positive infections were identified using a two stage procedure, first, viral RNA purification was conducted using either of three commercial kits from QIAGEN (Manchester, UK); QIAamp Viral RNA Mini Kit (Catalogue # 52906), RNeasy ® QIAcube ® HT Kit (Catalogue # 74171) and QIASYMPHONY ® RNA Kit (Catalogue # 931636) followed by real-time reverse-transcription PCR (RT-PCR) using one or two of four SARS-CoV-2 detection assays we deployed at the centre, since the beginning of the pandemic namely the Berlin Charite protocol, Europe Virus Archive Global (EVA-g) protocol, Beijing Genomics Institute (BGI) protocol and Da An Commercial Kit protocol[18]. RT-PCR was undertaken using primer/probes from the four protocols; (i) the Berlin (Charité)targeting E i.e., envelope gene, N i.e., nucleocapsid gene or RdRp i.e., RNA-dependent RNA-polymerase gene), (ii) European Virus Archive—GLOBAL (EVA-g) (targeting E or RdRp genes), (iii) Da An Gene Co. detection Kit (targeting N or ORF1ab) and Beijing Genomic Institute (BGI) RT-PCR kit (targeting ORF1ab).

**SARS-CoV-2 genome amplification, library preparation and sequencing**. At the beginning of the epidemic all SAR-CoV-2 positive samples were processed for whole genome sequencing using the ARTIC nCoV-2019 sequencing V.1 protocol[19]. Later we revised this to include only samples with real time PCR cycle threshold (Ct) score below 35.0. Viral RNA was extracted from 140 μl of NP/OP swabs using the QIAamp Viral RNA Mini Kit (Qiagen, Manchester, UK) according to the manufacturer's instructions except that carrier RNA was excluded at the lysis step. The purified viral RNA was then titrated based on a dilution factor that was determined from the relative amount of virus material in a sample as determined by Ct score. A reverse transcription (RT-PCR) reaction was carried out using the multiplex ARTIC primer-pools A and B (Supplementary Table 5). The reaction volume was carried out at half the recommended amount from the ARTIC sequencing protocol V.1[19] and the PCR thermocycling reactions were run for 40 cycles. Primer pool A and B PCR products were pooled together to make a total of 25 μl and cleaned using 1× AMPure XP beads (Beckman Coulter). The pellet was resuspended in 15 μl nuclease-free water and 1 μl of the eluted sample was quantified using the Qubit dsDNA HS Assay Kit (ThermoFisher). End-prep reaction was performed according to the ARTIC nCoV-2019 sequencing protocol with 200 fmol (50 ng) of cDNA and the NEBNext Ultra II End repair/dA-tailing kits and incubated at 20 °C for 5 min and 65 °C for 5 min. From this, 1.5 μl of DNA after End-Prep was used for native barcode ligation using NEBNext Ultra II Ligation (E7595L). In the absence of the Ligation Module, this step was performed using 10.5 μl of Blunt/TA Ligase Master Mix (M0367L). Incubation was performed at 20 °C for 20 min and at 65 °C for 10 min. Barcoded samples were pooled together. The pooled and barcoded DNA samples were cleaned using 0.4× AMPure XP beads followed by two ethanol (80%) washes and eluted in 35 μl of nuclease free water. Adaptor ligation was performed using 50 ng of the pooled sample, NEBNext Quick Ligation Module reagents (E6056L) and Adaptor Mix II (ONT) and incubated at room temperature for 20 min. Final clean-up was performed using 0.4× AMPure XP beads and 12 μl of Short Fragment Buffer (ONT). The library was eluted in 15 μl Elution Buffer. The final library was normalised to 15 ng prior to loading on a flow-cell and sequencing on a MinION Mk1B device.

**Genome assembly**. We used the ARTIC bioinformatics protocol to generate consensus sequences by aligning sequenced amplicons from each sample to the reference genome (Genbank accession MN908947.3). In brief, raw FAST5 files were converted to fastq format using the high accuracy basecalling mode and reads corresponding to each barcode were binned together (demultiplexed) based on strict parameters using ONT Guppy v4.0.11. The reads were then filtered based on length (300 ≤ X ≤ 700) and low-quality reads (phred score ≤7) were dropped. Consensus genomes were generated from reads corresponding to each barcode by aligning the reads from each sample to reference SAR-CoV-2 genome (Genbank accession MN908947.3). The consensus genomes were polished using raw signals using nanopolish[20,21] and positions with insufficient genome coverage (coverage <20 reads) were masked with the IUPAC ambiguous bases (N).

**SARS-CoV-2 lineage and clade assignment**. Consensus sequences were assigned lineages using the Pangolin toolkit (version 2.3.2) with pangoLEARN (version 2021-02-10). Phylogenetic clades were assigned using NextClade (version 0.13.0).

**Sampling from global dataset for context genomes**. SARS-CoV-2 whole genome sequences were downloaded from the Global Initiative on Sharing All Influenza Data (GISAID) database (https://www.gisaid.org/) as at 28th February 2021. Sequences collected later than 31st July 2020 were removed. Then, incomplete genomes (defined by length <29,500 nucleotides) and those with incomplete information on the date of collection were removed. The sequences were grouped by country and month. Five sequences from each group (i.e., for each month for each country) were sampled using the Augur filter command to obtain a final set of 2252 global sequences.

**Alignment and phylogenetics**. We used a modified NextStrain[22] Augur based pipeline to perform quality control and phylogenetic analysis for a total of 406 sequences from the Kenyan coast. In brief, we removed any entry that was shorter than 24,000 nucleotides or with any missing metadata information such as date and then aligned using MAFFT[23] v.7.475(2020/Nov/23) against the reference sequence. Low quality and divergent sequences were removed from the alignment and the resulting alignment of 311 sequences was trimmed at the 5′ and 3′ regions. Three known homoplasmic positions (13,402, 24,389 and 24,390) were masked in the alignment. A maximum likelihood tree was created using IQTree[24] version 2.0.3. A time resolved tree was inferred using TreeTime[25] version 0.81 with a clock rate set to 0.0008 and standard deviation 0.0004 under a skyline coalescent model and rooted using the reference genome (Wuhan-Hu/2019).

**Estimating importation events**. We used two approaches to infer the approximate number of introductions in the regions. The first approach was based on observation of fresh lineages during the early phase of the epidemic together with available epidemiological data. We assumed that a lineage comprised an introduction if it was observed early (March–30th April 2020) in the epidemic. We deduced that a sequence was an introduction if the sample was collected from an individual with a history of travel in the last 14 days or if the sample was collect at a point of entry (PoE).

We applied two approaches for computational phylogeographic estimation of ancestral state of internal nodes using the geographic traits of sequence tips from a total of 311 focal (locally collected sequences) that were selected in the previous phylogenetic step with global contextual sequences ($n = 2077$). This approach was run using the ancestral trait reconstruction approach implemented with the Augur-based pipeline. Ancestral trait reconstruction was conducted based on discrete parameters to infer the origin of ancestral nodes. The information was encoded and stored in a json file suitable for visualisation on auspice web-based tool.

In addition, a geographical trait file was generated by grouping sequences as either Kenyan or belonging to a broader geographical regions (Africa, Europe, North and South America, Asia or Oceania). This traits files was used together with a maximum-likelihood tree inferred in the phylogenetic step as used input into pastML[26], a fast ancestral reconstruction tool that implements maximum likelihood based ancestral reconstruction (ACR) methods. ACR was conducted using a marginal probabilities approximation (MPPA) with an F81-like transition model between state using two sets of contextual sequences i) one that used fewer number of sequences and ii) one that used a broader number of contextual sequence ($n = 2077$). We made used of both country specific traits and grouping of countries into broader geographical regions to increase the signal and reduce uncertainty amongst internal nodes close to the root.

Statistical analyses were performed using R version 4.0.2 and are described in the figure legends.

**Reporting summary**. Further information on research design is available in the Nature Research Reporting Summary linked to this article.

## Data availability

This study did not generate unique reagents, but raw data and code generated as part of this research can be found in the Supplemental files as well as on public resources as specified in the Data and code availability section below. SARS-CoV-2 sequence data used in this analysis are publicly available from GenBank accession numbers (MW751078-MW751422, MW931663-MW931714). A summary json file for interactive phylogenetic analysis is available as Supplementary Data 1 and can be loaded and visualised using Auspice web-based tool.

## Code availability

Source code and pipeline output intermediate files that was used in the analysis of this work are available from GitHub (https://github.com/george-githinji/sars-cov-2-early-phase-manuscript) and data used to create the figures can also be found in the supplemental files.

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

## Acknowledgements

The authors would like to thank members of the county rapid response teams who collected nasal and oral swabs, and all the frontline workers at the health and testing facilities. Special thanks to the COVID-19 testing team at KWTRP who conducted the laboratory PCR assays. We are grateful to those who have shared genome sequence data on GISAID (https://www.gisaid.org/). The full GISAID acknowledgment file is available in Supplementary Data 2. We thank the ARTIC Network (https://artic.network/) for providing us with primers and technical input and teams from the Oxford Nanopore Technology for the technical and material support. We would also like to acknowledge Prof La Scola Bernard for providing us with a heat-inactivated SARS-CoV-2 positive control that were used to validate our sequencing assays. This work was supported by the National Institute for Health Research (NIHR) (project references 17/63/82 and 16/136/33 using UK aid from the UK Government to support global health research, The UK Foreign, Commonwealth and Development Office and Wellcome Trust (grant# 102975; 220985) G.G. is funded and supported by NIHR funded GeMVi and TIBA projects (Grant number 17/63/82 and 16/136/33). C.A. is funded by the Initiative to Develop African Research Leaders (IDeAL) through the DELTAS Africa Initiative [DEL-15-003]. The DELTAS Africa Initiative is an independent funding scheme of the African Academy of Sciences (AAS)'s Alliance for Accelerating Excellence in Science in Africa (AESA) and supported by the New Partnership for Africa's Development Planning and Coordinating Agency (NEPAD Agency) with funding from the Wellcome Trust [107769/Z/10/Z] and the UK government. The views expressed in this publication are those of the authors and not necessarily those of AAS, NEPAD Agency, Wellcome Trust NIHR or the Department of Health and Social Care or the UK government. This was submitted for publication with permission from Director of KEMRI.

## Author contributions

G.G., D.J.N., C.A., I.L.O., G.W., S.M.K., A.A. and P.B. Conceptualised and designed the study. G.G., K.S.M., Z.R.L., D.O.O. and J.M.M. performed sequencing and provided laboratory support. C.A., G.G. and D.J.N. Provided phylogenetic analysis. B.T., I.L.O., A.A., E.B., C.A., K.K., R.A., J.K., E.M., B.K., T.S., M.M., J.N., J.O. and B.S. provided administration and support for sample collection. G.G. and P.M.M. developed the spatial maps. E.O., G.G., C.A., S.K.M. and P.M.M. provided data curation and analysis. G.G., C.A., D.J.N., G.W., I.L.O., B.T. and P.B. wrote the original manuscript draft. All contributors wrote, reviewed and edited the final manuscript.

## Competing interests

D.J.N. is a member of the National COVID-19 Modelling Technical Committee, for the Ministry of Health, Government of Kenya. K.K., R.A. and J.K. are from the Ministry of Health, Government of Kenya. E.M., B.K., T.S., M.M., J.N., J.O. and B.S. are from the respective county departments of health. The other authors declare no competing interests.
