## [Peer Review File · Nature Communications]

REVIEWER COMMENTS

Reviewer #1 (Remarks to the Author):

Tracking the introduction and spread of SARS-CoV-2 in coastal Kenya G. Githinji et al Nat comms – reviewed 17 November 2020

Summary

The manuscript by Githinji et al gives a good outline of the introduction and spread of SARS-CoV-2 in Kenya. SARS-CoV-2 sequencing information from Africa is limited, probably due to resource limitations within the region therefore it is of high interest.

The manuscript is well written and outlines the challenges and successes in containing SARS-CoV-2 in Kenya. I applaud the authors for clearly outlining the limitations of their study in the discussion. I believe the manuscript would benefit from additional details and clarification about the methods used, please see comments outlined below.

Overall comment: Major revisions required

Reviewers Comments:

Line 91, page 4

Supplemental table 2: Could you add dates to this table to ease understanding of when the restrictions took effect?

Please spell out what KPA stands for also how was the mass testing rolled out:
Should this table be supplemental table 1 as it appears first in the manuscript?

Line 95, page 4

What is the study start date specifically?

Line 108, page 4

I am unable to determine the 'follow-up' cases from figure S1 are you referring to Supplemental figure 3

Line 120, page 5

Are the independent introductions defined by novel lineages not seen in the Kenyan dataset previously or travel history of cases?

Line 133, page 5

I believe this should read figure 3B

Please confirm the colouring of nodes in Figure 3B is correct. At the bottom of the phylogeny lineage A nodes (red nodes) are interspersed with Lineage B.1 nodes (purple nodes). This does not look correct as the SNPs that define these lineages are quite different and are usually separated on two distinct branches. Could this be an artefact of some genomes only being 80% complete and therefore missing informative regions of the genome?

Line 188, page 5

It is nice to see you have repeat samples collected from 19 different cases you discuss 3 cases that have a slightly different virus between the two samples. It would be nice to see the original and then

subsequent mutational profile of these specimens compared to your reference genome. There are potential genome coverage and bioinformatic quality issues that could have caused these differences as well as intra-host variability. It would be interesting to discuss these differences further in the manuscript.

Line 286, page 9

Please give specific names for the RNA extraction kits used as well as the primers and probes used for SARS-CoV-2 RT-PCR assays. It would be nice to understand if you obtained genomes from only the high load samples available in your positive sample set.

Line 294, page 9

Please outline the library preparation kit used

Line 297, page 9

Please outline your bioinformatic QC procedures clearly, what software did you use, what files did you manually inspect? This provides confidence to the reader and reviewer of the quality of your data. I could not find the supplemental txt 1, only a supplementary figure 1 and table 1.

Page 327, page 10

Please outline the temporal signal in your phylodynamic analysis using root-to-tip regression analysis including a coefficient (r^2).

Line 338, page 11

Please elaborate on the methods used to determine importation or local transmission, I do not understand how this was done in the methods described here.

General comments:

Please acknowledge the genomes used from the GISAID database in a supplementary file as per the format recommended on the GISAID website

Give details of the GISAID and fastq files generated by this study and the location they have been made public available.

References numbered 5 and 6 are the same

Reviewer #2 (Remarks to the Author):

Githinji and colleagues present a phylogenetic investigation into the introduction of SARS-CoV-2 in Kenya. This study produces many novel genotypes from a region of the world where genotypes were underreported and helps clarify the nature of what was then the leading of the pandemic in East Africa. The results and interpretation are fairly typical for this type of study. However, there is a lack of robustness in the description of the computational approach that make some of the results difficult to interpret. Further, I would like to see a deeper investigation into the geographic relationship among clusters post-introduction.

The inference of a mean root age of mid-2019 (Figure 4) quite old and is indicative of issues with the molecular clock analysis. Why was a UCLN clock used in BEAST instead of a strict clock and was any prior placed on the substitution rate? It seems likely that the inferred substitution rate is too slow to produce epidemiologically realistic results. Although unlikely in low divergence viruses like SARS-CoV-2, incorrect clock inference could bias phylogenetic inference.

The approach employed to estimate of the number of importations (Pages 10-11) too vague to be reliably assessed. Was this performed on the ML tree or the Bayesian inference? If it was the latter, this approach needs to be conducted over the posterior distribution of trees and not the summary tree. It seems more likely these counts arise from the ML tree, since the Bayesian analysis didn't include the global sequences. Either way, "manual inspection" (on Page 11) does not clarify this approach sufficiently. This ambiguity makes it difficult to fully evaluate the [potentially] most epidemiologically important result of this manuscript.

Minor Comments

It would be useful to see a display of the placement of Kenyan sequences relative to other regions of the world represented by the 983 additional genome sequences (which is not apparent in Figure 4).

Table 2 is slightly confusing. I don't understand how the "Entry Source" and "Comments" can describe several different introductions (reportedly 45 introductions in the case of B.1.1). Also, "Proportion of sequences" should be "Percentage of Sequences", since these values exceed 1.

The use of citations throughout this manuscript is far too conservative. As an example "Similar investigations in Iceland, 79 Singapore and Europe have been undertaken" has no citations. Also, none of the phylogenetic approaches or tools are cited.

The near-duplication of the portions of the methods sections in "Methods" and "Methods Details" is more confusing than helpful. I would request that the authors remove the redundant portions of these sections.

General response

Thank you very much for taking your time to review our manuscript. We are grateful and value your comments. We have addressed each of the comments in turn below:

Reviewer #1

Summary

The manuscript by Githinji et al gives a good outline of the introduction and spread of SARS-CoV-2 in Kenya. SARS-CoV-2 sequencing information from Africa is limited, probably due to resource limitations within the region therefore it is of high interest. The manuscript is well written and outlines the challenges and successes in containing SARS-CoV-2 in Kenya. I applaud the authors for clearly outlining the limitations of their study in the discussion. I believe the manuscript would benefit from additional details and clarification about the methods used, please see comments outlined below.

Overall comment: Major revisions required

Our response: Indeed we have undertaken a major revision of the paper as suggested. We have undertaken a reanalysis and revised the text, figures and tables accordingly. We have included additional sequence data (n = 37). We revised our data processing pipeline to implement the augur pipeline (with modifications). We also provide our analysis in augur json format to facilitate sharing, review and publication of our work.

Line 91, page 4

Supplemental table 2: Could you add dates to this table to ease understanding of when the restrictions took effect. Please spell out what KPA stands for also how was the mass testing rolled out: Should this table be supplemental table 1 as it appears first in the manuscript?

Our response:

Thank you for the recommendation. We have added the dates and spelt out what KPA stands for.

- a) KPA stands for Kenya Ports Authority and is given in full now.
- b) The Ministry of Health referred to Mass testing but in practice what was meant was “targeted large-scale testing”. We have corrected this.
- c) Indeed, this should be Supplemental table 1. We have corrected the text to reflect the change.

Line 95, page 4

What does the study start date specifically?

Our response:

The Rapid Response Teams (RRTs) began collecting samples for SARS-CoV-2 detection from 12th March 2021. However, the first PCR positive sample was obtained on March 17th, 2020 and tested on the 21st March 2020. This is indicated in the revised manuscript.

Line 108, page 4

I am unable to determine the ‘follow-up’ cases from figure S1 are you referring to Supplemental figure 3

Our response:

Indeed, we are referring to supplemental figure 3. We have since revised the figure as part of the requested major revision to provide additional information and data.

Line 120, page 5

Are the independent introductions defined by novel lineages not seen in the Kenyan dataset previously or travel history of cases?

Our response:

In the early phase of the epidemic (March, April and May 2020), we defined independent introductions by novel lineages that we had not previously seen in the Kenyan dataset and by travel history of the cases to identify importations. For example., we inferred that a sequence was an introduction if the sample was collected from an individual with a history of travel in the last 14 days or if the sample was collect at a point of entry (PoE). Thus we counted at least 4 lineage A introductions between 30th March and 13th April but an overall of 15 introductions. A subtree of this lineage is provided in supplementary figure 7A and we updated table 2 to reflect this update. Where travel information was available to indicate persons arriving in the country carrying the same lineage, we counted these as separate introductions, as shown in table 2. We have since clarified this approach in the methods from line 211-238 by adding a new subsection titled “estimating importation events” This section describes a computational ancestral reconstruction approach to support our findings. We also discuss the pros and cons of this approach in the discussion section from line 569.

Line 133, page 5

I believe this should read figure 3B

Our response:

Yes indeed this should have read figure 3B in the previous submitted manuscript, although in the revised manuscript we now include a correct reference to figure 3A.

Please confirm the colouring of nodes in Figure 3B is correct. At the bottom of the phylogeny lineage A node (red nodes) is interspersed with Lineage B.1 nodes (purple nodes). This does not look correct as the SNPs that define these lineages are quite different and are usually separated on two distinct branches. Could this be an artefact of some genomes only being 80% complete and therefore, missing informative regions of the genome?

Our response:

As noted by the reviewer, in the previous figure 3B, Lineage A appeared to have been interspersed with lineage B.1 nodes. On re-analysis we found that this was not correct and was an artefact of the figure drawing and labelling. We have corrected this figure in the revised manuscript by annotating the sequence metadata with the correct lineage assignment and presented it as figure 4 in the manuscript. We have adopted a new node colour pallete in the revised manuscript partly because the improved lineage assignment toolkit provided a lot more lineages that in the previous analysis. Importantly, sequences in this cluster are lineage A and A.1 which is consistent with the SNPS that define these lineages.

Line 188, page 5

It is nice to see you have repeat samples collected from 19 different cases you discuss 3 cases that have a slightly different virus between the two samples. It would be nice to see the original and then subsequent mutational profile of these specimens compared to your reference genome. There are potential genome coverage and bioinformatic quality issues that could have caused these differences as well as intra-host variability. It would be interesting to discuss these differences further in the manuscript.

Our response:

We undertook a re-analysis to check on sequence quality and found 2 of the genomes failed QC because because they were shorter than 24,000 nucleotides long. We have removed them, leaving 17 repeats. For the 4 repeats specifically referred to we have added the profile of variation among the original and the mutation profile of the specimens as highlighter plots (in supplementary figures 2). In addition we provide a detailed outline of the differences in the results section of the revised manuscript. In summary we find that the discordance could be explained by missing sequence information but for one of the individuals R_5, we find striking differences among the sequences raising potential for a super infection or sampling error during this one sample collection.

Line 286, page 9

Please give specific names for the RNA extraction kits used as well as the primers and probes used for SARS-CoV-2 RT-PCR assays. It would be nice to understand if you obtained genomes from only the high load samples available in your positive sample set.

Our response:

We have updated the methods section of the revised manuscript and described the kits and assays in more details. In brief we used the following:

Extraction kits: viral RNA purification from the raw samples was extracted using either of three commercial kits from QIAGEN (Manchester, UK); QIAamp Viral RNA Mini Kit (Catalogue # 52906), RNeasy[®] QIAcube[®] HT Kit (Catalogue # 74171) and QIASYMPHONY[®] RNA Kit (Catalogue # 931636):

RT-PCR was undertaken using primer/probes from the following four protocols, the details of which we described elsewhere¹⁹; (i) the Berlin (Charité) targeting E i.e. envelope gene, N i.e. nucleocapsid gene or RdRp i.e. RNA-dependent RNA-polymerase gene), (ii) European Virus Archive – GLOBAL (EVA-g) (targeting E or RdRp genes), (iii) Da An Gene Co. detection Kit (targeting N or ORF1ab) and Beijing Genomic Institute (BGI) RT-PCR kit (targeting ORF1ab).

We obtained genomes from samples with a wide range (Ct score 15 -35) of Ct values (i.e. viral load) We have added a plot of the length of the genome against the corresponding Ct score and a regression line that shows a negative correlation between the recovered genome and the viral load. Genomes from low viral load samples were often incomplete. We have added supplementary figure to show the relationship between viral load and genome coverage.

Line 294, page 9

Please outline the library preparation kit used

Our response:

The Minion library preparation steps are outlined in the methods section (page 5 line143-169) of the revised manuscript as follows; “1.5µl of DNA after End-Prep was used for native barcode ligation using NEBNext Ultra II Ligation (E7595L). In the absence of the Ligation Module, this step was performed using 10.5µl of Blunt/TA Ligase Master Mix (M0367L). Incubation was performed at 20°C for 20 minutes and at 65°C for 10 minutes. Barcoded samples were pooled together. The pooled and barcoded DNA samples were cleaned using 0.4X AMPure XP beads followed by two ethanol (80%) washes and eluted in 35µl of nuclease free water. Adapter ligation was performed using 50ng of the pooled sample, NEBNext Quick Ligation Module reagents (E6056L) and Adapter Mix II (ONT) and incubated at room temperature for 20 minutes. Final clean-up was performed using 0.4X AMPure XP beads and 12 µl of Short Fragment Buffer (ONT) and the library was eluted in 15µl Elution Buffer”.

Line 297, page 9

Please outline your bioinformatic QC procedures clearly, what software did you use, what files did you manually inspect? This provides confidence to the reader and reviewer of the quality of your data. I could not find the supplemental txt 1, only a supplementary figure 1 and table 1.

Our response:

Many thanks for this suggestion. We have outlined the bioinformatics procedure including quality controls at various stages in the methods section (page 6 line 172). For genome assembly we use the artic bioinformatics protocol and tool to align the amplicons to a reference genome. The protocol only considers amplicons that are between 300 and 700 base-pairs long (, sequences with a mean low quality score (phred <7) were dropped. The next quality control was implemented during the phylogenetics step and outlined in the alignment and phylogenetics subsection page 7 line 198 of the methods section.

Page 327, page 10

Please outline the temporal signal in your phylodynamic analysis using root-to-tip regression analysis including a coefficient (r2).

Our response:

The temporal signal was obtained using root-to-tip regression and the result included in supplementary figure 2. The coefficient of regression based on 311 sequences in the phylogeny was 0.33, which suggests minimal molecular-clock-like accumulation of mutations in the sequenced samples. We obtained a substitution rate estimate of 17.695 substitutions per year in the context of the reference sequence. We think that this arises due to the multiple introductions of divergent lineages during this period (weeks/months).

Line 338, page 11

Please elaborate on the methods used to determine importation or local transmission, I do not understand how this was done in the methods described here.

Our response:

This is an important issue needing an improved explanation. We have provided an elaboration of the text to clarify this in the methods and “Estimating importation events” section on page 7 of the revised manuscript. This includes a new subsection in the methods section titled “Estimating importation events”.

We assume that a lineage comprises an introduction if we observed it early (March –May) in the epidemic and if the individual had a history of travel in the last 14 days or the case was detected at a point of entry. For example we consider a new lineage in March and April as an introduction. However, this approach loses resolution on the exact number of introductions when we observe multiple sequences with the same lineage and with no travel history. If sequences clustered more closely with other global sequence data, it could be considered as a separate introduction – this implies that the choice of the number and sampling criteria for selecting contextual sequences from the global dataset is important and influences the number of introductions that we can count.

This approach is less applicable as the epidemic spread further in the community, (i.e. May-July), when a signal based on count of lineages to infer introductions gets blurred.

Furthermore, we know that the sequenced sample is only a small proportion of the population and therefore the ability to estimate the exact number of introductions from outside the country is a

challenge. Our aim was to estimate the number of potential introduction,. We augment our approach by incorporating an established phylogeographic method using a maximum likelihood ancestral reconstruction approach (Ishikawa et al 2019) implemented in pastML toolkit assuming an F81 model. We do this using two datasets, a small dataset of 742 global sequences selected based on proximity scoring of coastal sequences (supplementary figure 5A) and an expanded dataset of 2,252 global sequences (supplementary figure 5B).

General comments:

Please acknowledge the genomes used from the GISAID database in a supplementary file as per the format recommended on the GISAID website. Give details of the GISAID and fastq files generated by this study and the location they have been made public available.

Our response:

This has been done and attached in the supplementary materials. We provide all the GISAID based accession number as well as the Genbank assigned accession number for the submitted sequences.

References numbered 5 and 6 are the same

Our response:

Thank you for notifying us. The extra reference has been removed.

Reviewer #2 (Remarks to the Author):

Githinji and colleagues present a phylogenetic investigation into the introduction of SARS-CoV-2 in Kenya. This study produces many novel genotypes from a region of the world where genotypes were underreported and helps clarify the nature of what was then the leading of the pandemic in East Africa. The results and interpretation are fairly typical for this type of study.

Our response:

We are grateful for the comments and we have addressed your comments and suggestions as outlined below:

- a. ... there is a lack of robustness in the description of the computational approach that make some of the results difficult to interpret.**

Our response:

We have added a thorough description of the computational workflow including genome assembly and phylogenetics (on Page 7 paragraph 3). In addition, we improved the phylogenetics analysis by incorporating additional tools and parameters and this is described on page N paragraph Y. To summarise this approach, we performed a maximum likelihood phylogenetic analysis using over 2000 contextual sequences (from where/ selected how?), we then used the output maximum likelihood tree to infer a time-resolved phylogenetic tree using a coalescent model with Timetree. Finally, we performed ancestral reconstruction using a computational rigorous approach implemented in pastML. This infers the number of introductions through a more robust procedure.

- b. Further, I would like to see a deeper investigation into the geographic relationship among clusters post-introduction**

Our response:

Data were limited for this, but we have attempted further geographic investigation by extending the period of analysis from June to 31st July and have added 37 new genomes in the phylogenetic analysis. In addition we provide details of the circulating lineage B.1 in Mombasa post introduction on the section on “circulating lineages between March and July 2020” in the revised text. For example we note the introduction of B.1.33 lineage in Mombasa, that potentially seeded the infection cluster in Lamu county post introduction into Kenya. (page 9 line 297-302).

“Lineage B.1.33 was detected in Lamu and comprised the earliest number of cases from Lamu. (n=16). Two cases were reported in Mombasa from samples collected on 11th May 2020 and 29th June 2020 respectively. The sample collected on 11th May 2020 clustered separately from the rest of the samples and could have comprised a separate introduction in Mombasa. The sample collected from Mombasa on the 29th June 2020 was part of a phylogenetic clade with the Lamu samples. Analysis of the ancestral nodes provide evidence for an ongoing epidemic in Lamu prior to the date of the first sequenced sample. Lamu is the only county we observed an epidemic from a SARS-CoV-2 lineage associated with sequences from South America. (Supplementary figure 5B and 7).”

We observe something similar with introduction of lineage A cases that did result in widespread introduction although they were identified in multiple counties in the coast (Mombasa, Kwale and Taita Taveta).

“Lineage A was the second most frequent lineage from the full sequenced dataset (n=406) particularly among individuals sampled at the ports of entry (PoE) based in Kwale and Taita

Taveta counties (Figure 3, Table 2) or from individuals with a history of travel (n=8) to Tanzania (Table 1). We described at least 15 independent introductions of this lineage into the coast region (Table 2, supplementary figure 6). The first lineage A sample was detected in Mombasa from samples collected on the 4th of April (C314 – detected at the border and C293 – travel associated). These two samples and in addition to sample C571, differed by 1 mutation from the reference and formed a single phylogenetic clade whose ancestral date was inferred as 26th March 2020 (date confidence interval 18th March – 1st April 2020). Samples C7603, C7605, C7866, and C14075 also formed a single cluster but comprised of multiple introduction given that they were detected at the Lunga Lunga PoE and samples collected on different time points (supplementary figure 7). Two additional samples collected on the 14th May 2020 support a common source of infection but comprised 2 separate introductions. Our ancestral reconstruction provides evidence that these multiple introductions of lineage A could be of Asian origin (supplementary figure 7). We identified a single lineage A.1 sample which could be an indicator for transmission of this lineage in the region. The paucity of genomic data from the region limits our description and understanding of the local dynamics of transmission.”

We plan to take this analysis further in a second paper.

c. The inference of a mean root age of mid-2019 (Figure 4) quite old and is indicative of issues with the molecular clock analysis

Our response:

We undertook a re-analysis as follows: (a) We increased our dataset by sequencing additional samples collected in June and July 2020 (n=37) (b) We removed sequences with poor genome coverage i.e, sequences with more than 6,000 ambiguous bases. We then generated a root-tip divergence plot based on the maximum likelihood tree and inferred a robust and accurate mean root age of late 2019 to early 2020. This plot is shown in supplementary figure 2A.

d. Why was a UCLN clock used in BEAST instead of a strict clock and was any prior placed on the substitution rate? It seems likely that the inferred substitution rate is too slow to produce epidemiologically realistic results. Although unlikely in low divergence viruses like SARSCoV-2, incorrect clock inference could bias phylogenetic inference.

Our response:

Thank you for raising this concern. Given the lack of good phylogenetic temporal signal based on the earlier sequence set, and the evidence for slow substitution rate in SARS-CoV-2, we recognize that the relaxed clock was a mistake. We have since revised this section as follows: (a) We apply a strict criteria for sequence inclusion in the phylogenetic analysis (>80% genome coverage) (b) We diagnose the alignment and remove sequences that were either too short and divergent. In addition we mask 3 homoplasmic regions. (c) We then infer a maximum likelihood tree using IQtree and use the output maximum likelihood tree as input to infer a time-resolved maximum likelihood tree using TreeTime and assume a skyline coalescent model with a clock rate set to 0.0008 and standard deviation 0.0004. (d) the tree is then rooted using the reference genome Wuhan/Hu-1/2019.

e. The approach employed to estimate of the number of importations (Pages 10-11) to too vague to be reliably assessed. Was this performed on the ML tree or the Bayesian inference? If it was the latter, this approach needs to be conducted over

the posterior distribution of trees and not the summary tree. It seems more likely these counts arise from the ML tree, since the Bayesian analysis didn't include the global sequences. Either way, "manual inspection" (on Page 11) does not clarify this approach sufficiently. This ambiguity makes it difficult to fully evaluate the [potentially] most epidemiologically important result of this manuscript.

Our response:

This is indeed an important issue and was also raised by first reviewer. We agree manual inspection is not sufficiently robust, and we have therefore undertaken a rigorous analysis as defined in the response in section **"Line 338, page 11 Please elaborate on the methods used to determine importation or local transmission, I do not understand how this was done in the methods described here"**. We added a new section in the methods section of the revised manuscript titled "Estimating importation events" page 7 line 212-238 of the revised text. We provide a detailed description of applying this approach to infer the introduction of lineage A cases in coastal region in the results section page 9 line 347. Although the approach may have some challenges, some of which we outline on page 13 in the discussion section of the revised manuscript line 567.

Minor Comments

It would be useful to see a display of the placement of Kenyan sequences relative to other regions of the world represented by the 983 additional genome sequences (which is not apparent in Figure 4). Table 2 is slightly confusing.

Our response:

We have improved on this figure and produced a higher resolution supplementary figure 4 that shows the placement of Kenyan sequences relative to other regions of the world. In addition, we provide an augur json file that can be visualised in auspice. This file contains our dataset and can be dynamically viewed.

I don't understand how the "Entry Source" and "Comments" can describe several different introductions (reportedly 45 introductions in the case of B.1.1).

Our response:

We agree this was confusing and we have revised this section and table to clarify what we meant. In brief we removed the word "entry" as the column header and replaced with "source" to clarify that we intended to refer to the activity through which the case was identified. For example, "border" refers to cases identified at port of entry and "local" refers to cases with no history of travel. The "comments" column captures additional information that was collected by the county RRT teams and could inform on whether the case was associated with travel and if so where from.

Based on the updated pangolin toolkit and that has seen several updates leading to version 2.4.2. that we used here, we did not obtain lineage B.1.1 as previously suggested and the sequences are now classified into B.1.1 sublineages.

Also, "Proportion of sequences" should be "Percentage of Sequences", since these values exceed 1.

Our response:

This has now been corrected and the column labelled as percentage of sequences.

The use of citations throughout this manuscript is far too conservative. As an example, “Similar investigations in Iceland, 79 Singapore and Europe have been undertaken” has no citations. Also, none of the phylogenetic approaches or tools are cited.

Our response:

This has been corrected. We have updated the citations and provided relevant references for all the phylogenetic approaches or tools used in the analysis as well as the version numbers.

The near duplication of the portions of the methods sections in “Methods” and Methods Details” is more confusing than helpful. I would request that the authors remove the redundant portions of these sections.

Our response:

The duplicated sections have been removed and we only have a single methods section starting from line 89.

In addition to the reviewers responses above we included the augur json files(augur_json.zip) in the supplementary material suitable for reviewing of this work. Each of these files can be dragged and dropped in an auspice session for visualization.

REVIEWERS' COMMENTS

Reviewer #1 (Remarks to the Author):

Thank you to the authors for clarifying the manuscript thoroughly based on the reviewers initial comments.

I note there are still typographical and spelling errors throughout the text see below

I only have one major additional concern, there are no captions that explain the supplemental figures, yet they are very important and are referenced frequently in the main text. Please add captions to supplemental figures for ease of reference and to understand their importance to your analysis

The authors mention that travel history is only available for a subset of cases. It would be useful to make this clear in the Demographic characteristic section when you mention overall percentages without raw numbers. I note that this is contained within Table 1. But is not clear within the text. These unknown cases make up a significant (31.8%) of the studied cases and so should be outlined when citing percentages in the results.

Please enlarge the colours and labels on Supplementary figure 2. This figure is only mentioned in the discussion and not the results. It seems to outline your computational/bioinformatic approach to defining importation but in its current state it is unreadable.

Do the grey areas in supplementary figure 4 show missing positions?

It is difficult to see the colours in supplemental Figure 5

Extra spacing before citation in line 49, 52 and 62 but not 60 or 64. In some lines there is a space before the full stop after the citation and other not for example Line 82, 83 but not 73.

Extra 't' on line 84

Line 169 I believe should be supplementary figure 6

Line 193 Extra space after Fig. 3

Extra spaces in text line 201, 237, 238, 251, 299, 310

Missing spaces in text Line 203 between '.' and Sequence

Extra '.' In line 307

Check naming and capitalization of references to Figures and Supplemental files, multiple versions used throughout text

Reviewer #2 (Remarks to the Author):

Githinji and colleagues have provided a substantially improved manuscript. I have some further comments below.

I am glad to see that the authors incorporated additional contemporaneous genomes to contextualize the Kenyan epidemic. However, the lack of sequenced genomes from other counties in sub-Saharan Africa complicates the inference of locations of introduction into Kenya. I admit that this lack of genomes is not the fault or responsibility of the authors. However, given the number of documented COVID-19 cases stopped at the land border, it is highly likely that virus was also introduced this way. And if few reference genomes from neighboring countries were available (and epidemics in these countries were also seeded by virus from Europe), their provenance can be difficult to establish. Could a formal analysis of the 67 genomes acquired at border crossings (line 282) be used to inform this issue? Regardless, this limitation should be noted/highlighted.

It would be helpful to see trees/clusters colored by ancestral state reconstruction in a Figure (or Supplementary Figure). The auspice download is a nice touch, but one not readily accessible to the casual reader.

There appear to be mis-numbering in the reference to Supplementary Figures (e.g., S7 at lines 153 and 166). Also, I don't see Legends for the Supplementary Figures.

Line 33, should 2021 be 2020?

Lines 33-34, not a complete sentence. Also, are the authors suggesting that 'all' introductions were of European origins, or just those they could confidently assign? Clearly there is transmission across the Kenya-Tanzania border, as evidenced by the screening.

General Response

Thank you very much for taking time to review the manuscript and delighted for its acceptance for publication. We have addressed the reviewers comments in turn below.

Reviewer #1 (Remarks to the Author):

Thank you to the authors for clarifying the manuscript thoroughly based on the reviewers' initial comments.

I note there are still typographical and spelling errors throughout the text see below

I only have one major additional concern, there are no captions that explain the supplemental figures, yet they are very important and are referenced frequently in the main text. Please add captions to supplemental figures for ease of reference and to understand their importance to your analysis

Our response:

This is noted. Captions for the supplemental figures have now been added for ease of reference.

The authors mention that travel history is only available for a subset of cases. It would be useful to make this clear in the Demographic characteristic section when you mention overall percentages without raw numbers. I note that this is contained within Table 1. But is not clear within the text. These unknown cases make up a significant (31.8%) of the studied cases and so should be outlined when citing percentages in the results.

Our response:

Thank you for pointing out this. This has now been clarified in the main text demographics characteristics section line 120.

Please enlarge the colours and labels on Supplementary figure 2. This figure is only mentioned in the discussion and not the results. It seems to outline your computational/bioinformatic approach to defining importation but in its current state it is unreadable.

Our response:

The figure is important and refers to the findings of a specific lineage in Lamu similar to B.1.1.33 that had been observed to circulate in South America. References to the figure has been made in the text line 238 and line 243 in the results section of the draft manuscript.

Do the grey areas in supplementary figure 4 show missing positions?

Our response:

Yes, indeed the grey areas in supplementary figure 4 show missing positions denoted as strings of N in the nucleotide sequence.

It is difficult to see the colours in supplemental Figure 5

Our response:

Supplementary Figure 5 had been increased slightly in size. The figure is a summary of global context of the Kenyan genomes (green tips) against the global sequences collected at the early during the same period (blue tips).

Extra spacing before citation in line 49, 52 and 62 but not 60 or 64. In some lines there is a space before the full stop after the citation and other not for example Line 82, 83 but not 73.

The extra spaces before citation in line 49, 52 and 62 and the extra full stops have now been removed.

Extra 't' on line 84

The extra t on line 84 has been removed.

Line 169 I believe should be supplementary figure 6

Yes, indeed this should be supplementary figure 6 and has been corrected (page 6 line 167)

Line 193 Extra space after Fig. 3

removed extra space after Fig. 3

Extra spaces in text line 201, 237, 238, 251, 299, 310. Missing spaces in text Line 203 between '.' and Sequence, Extra '.' In line 307

Our response:

Thank you for bringing these to our attention. We have provided the suggested edits to make the text consistent and grammatically correct.

Check naming and capitalization of references to Figures and Supplemental files, multiple versions used throughout text

Our response:

Thank you for highlighting this inconsistency. We have provided the required suggestion and rename all the figures so that they adhere to a single version.

Reviewer #2 (Remarks to the Author):

Githinji and colleagues have provided a substantially improved manuscript. I have some further comments below.

I am glad to see that the authors incorporated additional contemporaneous genomes to contextualize the Kenyan epidemic. However, the lack of sequenced genomes from other counties in sub-Saharan Africa complicates the inference of locations of introduction into

Kenya. I admit that this lack of genomes not the fault or responsibility of the authors. However, given the number of documented COVID-19 cases stopped at the land border, it is highly likely that virus was also introduced this way. And if few reference genomes from neighbouring countries were available (and epidemics in these countries were also seeded by virus from Europe), their provenance can be difficult to establish.

Could a formal analysis of the 67 genomes acquired at border crossings (line 282) be used to inform this issue? Regardless, this limitation should be noted/highlighted.

Our response:

Thank you for highlighting this observation. We added a formal analysis of these sequences and summarised in supplementary table 4. In addition, we describe our observation in line 173-179 and line 303-306 in the manuscript. We observe that lineage A and lineage B.1 could have been co-circulating in Tanzania at the same period. The lack of genomes from Tanzania is a major limit to our inferences. A detailed formal analysis would be important; however, we understand the current limitations of the data.

It would be helpful to see trees/clusters coloured by ancestral state reconstruction in a Figure (or Supplementary Figure). The auspice download is a nice touch, but one not readily accessible to the casual reader.

Our response:

Thank you for the suggestion, we have added supplementary figure 8 showing coloured by the ancestral reconstruction⁶

There appear to be misnumbering in the reference to Supplementary Figures (e.g., S7 at lines 153 and 166). Also, I don't see Legends for the Supplementary Figures.

Our response:

Thank you for pointing this omission. It has been corrected to refer to the correct figure supplementary figure 6 in this case.

Line 33, should 2021 be 2020?

Our response:

Yes indeed! This has been corrected to reflect the correct time period of 17th March to 31st July 2020.

Lines 33-34, not a complete sentence. Also, are the authors suggesting that 'all' introductions were of European origins, or just those they could confidently assign? Clearly there is transmission across the Kenya-Tanzania border, as evidenced by the screening

Our response:

We agree with the reviewer's comments and point of view. We have completed the sentence from line 33-35 on page 2 of the manuscript to reflect this point of view. In addition, a limited analysis of the 32 of the 67 samples collected that the border points we

asymptomatic of the lineage A and B.1. This means that prior to restriction of movement, several asymptomatic cases could have already transmitted across the border.